# Synthesis, Molecular Docking Study, and Cytotoxic Activity against MCF Cells of New Thiazole–Thiophene Scaffolds

**DOI:** 10.3390/molecules27144639

**Published:** 2022-07-20

**Authors:** Sobhi M. Gomha, Sayed M. Riyadh, Bader Huwaimel, Mohie E. M. Zayed, Magda H. Abdellattif

**Affiliations:** 1Department of Chemistry, Faculty of Science, Islamic University of Madinah, Madinah 42351, Saudi Arabia; 2Department of Chemistry, Faculty of Science, Cairo University, Giza 12613, Egypt; riyadh1993@hotmail.com; 3Department of Pharmaceutical Chemistry, College of Pharmacy, University of Ha’il, Ha’il 81442, Saudi Arabia; b.huwaimel@uoh.edu.sa; 4Department of Chemistry, Faculty of Science, King Abdulaziz University, Jeddah 21589, Saudi Arabia; mzayed@kau.edu.sa; 5Department of Chemistry, College of Science, P.O. Box 11099, Taif 21944, Saudi Arabia; m.hasan@tu.edu.sa

**Keywords:** thiophenes, thiazoles, multicomponent synthesis, hydrazonoyl halides, anticancer, molecular docking, toxicity radar, ADME

## Abstract

Investigating novel compounds that may be useful in designing new, less toxic, selective, and potent breast anticancer agents is still the main challenge for medicinal chemists. Thus, in the present work, acetylthiophene was used as a building block to synthesize a novel series of thiazole-bearing thiophene derivatives. The structures of the synthesized compounds were elucidated based on elemental analysis and spectral measurements. The cytotoxic activities of the synthesized compounds were evaluated against MCF-7 tumor cells and compared to a cisplatin reference drug, and against the LLC-Mk2 normal cell line using the MTT assay, and the results revealed promising activities for compounds **4b** and **13a**. The active compounds were subjected to molecular modeling using MOE 2019, the pharmacokinetics were studied using SwissADME, and a toxicity radar was obtained from the biological screening data. The results obtained from the computational studies supported the results obtained from the anticancer biological studies.

## 1. Introduction

Breast cancer affects 14% of all women living globally [1]. It is the most frequently diagnosed neoplasm in female patients. The essential treatments for breast cancer are surgery, radiotherapy, and chemotherapy, individually tailored to the patient. Unfortunately, one of the main problems in the pharmacotherapy of cancers, including breast cancer, is the rapid development of drug resistance. Thus, designing and synthesizing more efficient agents with fewer adverse effects is essential [2]. Investigating novel compounds that may be useful in designing new, less toxic, selective, and potent anticancer agents is still the main challenge for medicinal chemists. Several studies have been carried out using various sulfur heterocycles, including thiophene and thiazole, directed towards different pathologies. It is reported in the literature that compounds containing a thiophene core have attracted considerable interest in drug discovery due to their potential anticancer activity [3,4,5,6,7]. Many anticancer agents are available on the market that contain a thiophene nucleus, as shown in Figure 1, and these exhibit their activity via multiple pathways involved in cancer [8,9].

On the other hand, 1,3-thiazole derivatives have been extensively considered by researchers generating novel lead compounds and in drug development. Thiazole derivatives of thiosemicarbazone are the scaffolds of many natural, synthetic, and semi-synthetic drugs which exhibit numerous remarkable pharmacological activities, including antiparasitic, anti-inflammatory, and antineoplastic activities [10,11,12,13]. Furthermore, the thiazole ring is present in several anticancer drugs (Figure 1), including Tiazofurin, Dasatinib, Dabrafenib, Bleomycin, Ixabepilone, and Epothilone, making this skeleton an ideal candidate for the development of more potent and safer anticancer drugs [14,15,16].

Molecular hybridization is a beneficial approach to structural alteration involving the integration of a single species of two or more pharmacophores [11,17,18,19,20,21,22,23]. Over the last several years, hybrid drug design has been used as a prime method for developing novel anticancer therapies that can solve many of the pharmacokinetic disadvantages of traditional anticancer drugs. Thus, several studies have indicated that thiazole–thiophene hybrids have important anticancer activity [24].

Based on the above-mentioned promising aspects, and in continuation of our previous work to synthesise anticancer agents [11,24,25,26,27,28] from readily available, cheap, laboratory starting materials with anticipated biological activities, the strategy of this present work involved gathering the two bioactive entities, thiophene and thiazole, into one compact hybrid structure that may lead to enhanced anticancer activity due to the synergistic effect of both rings. Therefore, 2-acetylthiophene seemed to be a suitable starting material to fulfill this objective (Figure 2).

Apoptosis is essential to normal breast development and homeostasis. Pro- and anti-apoptotic signals are tightly regulated in normal breast epithelial cells. Dysregulation of this balance is required for breast tumorigenesis, and this increases the acquired resistance to treatments such as molecularly targeted therapies, radiation, and chemotherapies. Members of the pro- or anti-apoptotic BCL-2 family of proteins are key regulators of the apoptosis process, and 2W3L is one of the proteins in this family. Therefore, 2W3L is a promising target to improve the killing of breast cancer tumor cells [29,30]. The docking process was carried out by simulating the exchange of the most biologically active compounds, **4a,b**, **8a**, **11b**, and **13a,b**, with two types of breast cancer proteins (PDB = 2W3L). Moreover, ADME analysis showed that the compounds have drug-like properties. ProToxII is one of the most common tools for predicting pharmacokinetic drug toxicity [31].

## 2. Results and Discussion

### 2.1. Chemistry

Heterocyclic compounds with three components have been widely exploited to prepare substituted heterocyclic compounds [25,32,33]. Thus, combining 2-acetylthiophene (**1**) with thiocarbohydrazide (**2**) and *α*-keto hydrazonoyl chlorides (**3a–d**) in ethanol with a catalytic amount of triethylamine under refluxing conditions led to the formation of 4-methyl-5-(arylazo)-2-[(1-(thiophen-2-yl)ethylidene)hydrazineylidene]thiazol-3(2*H*)-amines (**4a–d**). Furthermore, this three-component process worked effectively with different substituent aryl groups on the hydrazonoyl chloride molecule (Figure 1).

We previously proposed the mechanistic pathway for this transformation starts with in situ condensations of 2-acetylthiophene (**1**) and thiocarbohydrazide (**2**) to give the respective hydrazone **5** [34]. Subsequent heterocyclization of intermediates **6a–d** gave the isolable products **4a–d** (Figure 1). Assignment of spectral data (via IR, NMR, MS, and elemental analysis) for the isolated products provided significant indications of their structures. For example, in the IR spectra of compounds **4a–d**, the appearance of characteristic bands at 3414–3420 and 3226–3230 cm^−1^ confirms the presence of the amino group, and in ^1^H-NMR, the singlet signal at approximately *δ* = 5 ppm is assigned to the resonance of amino protons on the thiazole ring [35].

Aspects of this methodology were extended to investigate the three-component reaction of 2-acetylthiophene (**1**), thiocarbohydrazide (**2**), and ethyl 2-chloro-2-(2-arylhydrazono)acetate (**7a,b**). Similarly, this process furnished the respective 3-amino-thiazolidine-4-one derivatives **8a,b**, as illustrated in Figure 2.

The structures of the isolated products **8a** and **8b** were established from the stretching vibration signals at 3428, 3340, 3235, and 1680 cm^−1^ in the IR spectra, which were attributed to NH_2,_ NH, and C=O groups, respectively. In addition, ^1^H-NMR revealed two singlet signals at *δ* = 5.08–5.11 and 10.76–11.24 ppm (D_2_O exchangeable) which were assigned to the NH_2_ and NH protons.

Compounds **3a** and **7a** were expeditiously transformed into **4a** and **8a**, respectively, via their reactions with *N*′-[1-(thiophen-2-yl)ethylidene]hydrazinecarbothiohydrazide (**5**) [34], as depicted in Figure 3.

To exploit the synthetic approach of a three-component system for the preparation of thiazole-3-amine derivatives, we condensed acetylthiophene (**1**) and thiocarbohydrazide (**2**) with either 2-bromo-1-arylethanone (**10a–d**) or *α*-chloro-dicarbonyl compounds (**12a,b**) under the previously employed conditions to obtain **11a****–d** or **13a,b**, respectively (Figure 4). Furthermore, the structures of the isolated products **11a–d** and **13a,b** were elucidated based on their spectral and analytical data (see Experimental Section).

### 2.2. Cytotoxic Potential

The cytotoxicity of the synthesized thiazoles **4a–d**, **8a,b**, **11a–d**, and **13a,b** was investigated against the human breast cancer MCF-7 and normal LLC-Mk2 cell lines using the MTT assay and cisplatin as the reference drug. The % inhibition was plotted against log concentration, with normalization applied and error bars for the MCF-7 cell line shown, as represented in Figure 3.

The results were used to plot a dose–response curve, from which the concentrations of the tested samples required to kill half of the cell population (IC_50_) were determined. In addition, cytotoxic activities were expressed as the mean IC_50_ calculated from three independent experiments. The results, represented in Table 1, Figure 3 and Figure 4, revealed that most of the tested compounds showed very variable activity compared to the reference drug.

Examination of the SAR leads to the following conclusions:The 1,3-thiazole derivatives **4b** and **13a** (IC_50_ = 10.2 ± 0.7 and 11.5 ± 0.8 μM, respectively) have promising antitumor activity against the breast carcinoma cell line (MCF-7), and showed greater activities than the cisplatin reference drug (IC_50_ = 13.3 ± 0.61 μM);The 1,3-thiazole derivatives **11c** and **11d** have poor antitumor activity (IC_50_ > 38 μM), while the rest of the evaluated thiazoles have moderate activity (IC_50_ = 13.6–23.7 μM);For 1,3-thiazoles **4**, **8**, and **11**: the introduction of an electron-donating group (eg. methyl group) into phenyl group at position 5 in the 1,3-thiazole ring enhances the antitumor activity, while the introduction of an electron-withdrawing group (chlorine) decreases the activity (**4b > 4a > 4c > 4d**; **8a > 8b**; and **11b >11a > 11c >11d**);For the substituent at position 5 of the 1,3-thiazoles: an acetyl group (Ac) gives higher activity than an ester group (CO_2_Et). **13a** (IC_50_ = 11.5 ± 0.7 μM) **> 13b** (IC_50_ = 16.3 ± 1.4 μM).

The effects of the cisplatin standard drug and the most active compounds, **4a**, **4b**, **8a**, **11b**, **13a**, and **13b**, against LLC-Mk2 (rhesus monkey normal kidney epithelial cells) were also measured, to produce a dose–response curve and to calculate the fifty percent cytotoxic concentration (CC_50_), as indicated in Table 1. The results showed that all examined compounds are non-toxic, because their CC_50_ toward normal cell lines was higher than 100 uM [36].

The selectivity index (SI) was calculated by dividing CC_50_ by IC_50_. Our results showed that most of the derivatives presented good selectivity index values, indicating higher potency than the cisplatin anticancer drug. When the test compounds were evaluated for their toxicity against normal cells, they exhibited low toxic effects, indicating the safe use of most of them, but this may require further in vivo and pharmacological studies.

### 2.3. Molecular Docking Studies 

Molecular docking is computational software routinely used for understanding the protein–receptor interaction with complexes. The docking process was carried out by simulating the exchange of the prepared compounds with two types of breast cancer proteins (**PDB = 2W3L**) [37] for compounds **4a,b**, **8a**, **11b**, and **13a,b** (Figure 5). Owing to the anticancer biological study results obtained, only these compounds were subjected to molecular docking because of their higher activity compared to the other synthesized compounds. Cisplatin energy was not calculated as it is difficult to calculate this using the software MOE2019, because it appears in the MOE system as a square planner molecule, not a cis molecule. To solve this issue, we used carboplatin as a reference docking drug, and the new results have been added to the corrected manuscript.

The docking score energies of compounds **4a**, **4b**, **8a**, **11b**, **13a**, **13b**, and **CarboPt** were (−5.911, −6.011, −6.161, −5.65, −5.436, −5.883, and −4.671 kcal/mol), respectively. These scores are only of moderate activity, not higher, and this can be explained by the following: only compound **4b** showed an interaction of the S(8) atom on the ligand to the receptor on the O of Arg 66 (B) by hydrogen donation, while all the other compounds had no measurable interaction, and all of them connected to the dummies by ligand exposure, as shown in Table 2. Having no measurable interactions depends on the exposure of the whole ligand to the protein. The results also indicated that compound **4b** was the best one, as its IC_50_ was closest to cisplatin, with good SI values of 46–30. These results are consistent with the data which show that **4b** was the only one with measurable interactions of high docking score energy.

The molecular docking studies were also carried out on HAS (Human Serum Albumin) PDB = 1AO6, but only with **4b**, **13a**, and carboplatin (Figure 6). This was to validate the results obtained from the docking studies on PDB=2W3l. The docking score energy of compound **4b** with PDB=1AO6 was −6.3 kcal/mol, while that of compound **13a** was −5.228 kcal/mol. Furthermore, both **4b** and **13a** showed no measurable interactions with 1AO6. The only interaction was via ligand exposure. In addition, the docking score of CarboPt with 1AO6 was −4.78 kcal/mol.

### 2.4. Toxicity Radar

The ProTox-II data showed that the tested compounds (**4a**, **4b**, **8a**, **11b**, **13a**, and **13b**) were predicted to have oral LD_50_ values ranging from 159 to 3000 mg/kg in a rat model, with (1 s, 4 s)-eucalyptol bearing the highest value, and quercetin holding the lowest one (Figure 7 and Table 3). Therefore, the SI (selectivity index) calculations for these compounds obtained from the biological studies were adequately compatible with the toxicity radar calculations, which validates the results obtained. Through screening the toxicity radar results, we found that compound **4b** had a higher predictable LD_50_, which agrees with the results obtained from the molecular docking and the biological activities. However, the prediction accuracy for all compounds was 12%, so these compounds should be further investigated and screened.

### 2.5. SwissADME Studies

ADME (absorption, distribution, metabolism, and excretion) studies, including drug-likeness analysis, are essential in drug discovery, and provide a reasonable decisiveness on whether or not inhibitors should be progressed to a biological system [31]. A potent antagonistic interaction of inhibitors with a receptor protein or enzyme can not guarantee the ability of an inhibitor to act as a drug; therefore, ADME assessment is essential in drug development. Inhibitors having low ADME properties and high toxicity effects on biological systems are often the dominant reasons for the failure of most medicines in the experimental phase.

Figure 8 shows the output of the ADME studies and the drug-likeness properties (refer to the Appendix A); it was observed that the **4b** and **13a** molecules display one or two violations of Lipinski’s rule, and the first violation is the molecular weight rule, with a result of 356.47–425.36 g/mol. The drug-likeness parameters are related to aqueous solubility and intestinal permeability, determining the first step of oral bioavailability [38]. The results also indicated good pharmacokinetic properties, in which compounds **4b** and **13a** have high gastrointestinal absorption.

### 2.6. Pred-hERG

Chemically similar compounds often bind to biologically diverse protein targets, and protein structures do not always recognize identical ligands. Pharmacological and off-target relationships between proteins and a ligand set help to improve machine learning confidence by interpolating the output prediction equalized by the compound similarity criteria. This pipeline helps to improve predictions of off-target drug effects, reducing false-negative errors. The Labmole server was used to predict Pred-hERG, and to predict similar compounds with structure–activity relationships (Table 4, Figure 8).

Chemical similarity is one of the most critical concepts in cheminformatics. One commonly used algorithm to calculate these similarity measures is the 2D Tanimoto algorithm employed here (Figure 9, Figure 10 and Figure 11). The resulting Tanimoto coefficient is fingerprint-based, encoding each molecule to a fingerprint “bit” position (MACCS), with each bit recording the presence (“1”) or absence (“0”) of a fragment of the molecule. Interpretation of the probability of toxicity for compounds **4b** and **13a** can be explained by the cytotoxicity diagram [31].

## 3. Experimental Section

### 3.1. Chemistry

#### 3.1.1. Experimental Instrumentation

All melting points were determined using electrothermal apparatus, and were left uncorrected. IR spectra were recorded (KBr disc method) using a Shimadzu FT-IR 8201 PC spectrophotometer. ^1^H NMR and ^13^C NMR spectra were recorded in DMSO solutions using a BRUKER 400 FT-NMR spectrometer, and chemical shifts were expressed in ppm using TMS as an internal reference. Mass spectra were recorded using a Shimadzu GC-MS QP1000 EX. Elemental analyses were carried out at the Microanalytical Center of Cairo University.

#### 3.1.2. General Procedure for Synthesizing the Thiazole Derivatives **4a–d**, **8a,b**, **11a–d**, and **13a,b**

A mixture of 2-acetylthiophene (**1**) (0.126 g, 1 mmol) and thiocarbohydrazide (**2**) (0.106 g, 1 mmol) in ethanol (20 mL) was refluxed with a few drops of hydrochloric acid for one hour. Then, without extraction of the hydrazone product, either the appropriate hydrazonoyl chlorides, **3a-d** or **7a,b,** or the *α*-halocarbonyl compounds, **10a****–****d** or **12a,b,** (1 mmol), were added with catalytic amounts of triethylamine, and the reaction mixture was refluxed for 4 h (monitored by TLC). Finally, the precipitate formed was isolated by filtration, washed with methanol, dried, and recrystallized from the appropriate solvent to give products **4a–d**, **8a,b**, **11a****–****d,** or **13a,b**, respectively. The physical properties and spectral data of the isolated products are listed below.

**4-Methyl-5-(phenyldiazenyl)-2-[((1-(thiophen-2-yl)ethylidene)hydrazineylidene]thiazol-3(2*H*)-amine (4a).** Red solid, 74% yield, m.p. 173–175 °C (EtOH); IR (KBr): *v* 3414, 3229 (NH_2_), 1602 (C=N) cm^−1^; ^1^H-NMR (DMSO-*d*_6_): *δ* = 2.25 (s, 3H, CH_3_-thiazole), 2.46 (s, 3H, CH_3_-C=N), 5.08 (s, 2H, Ar-H), 7.07–7.63 (m, 8H, Ar-H); ^13^C-NMR (DMSO-*d*_6_): *δ* = 13.21 (CH_3_-thiazole), 15.84 (CH_3_-C=N), 110.68, 125.11, 126.14, 127.39, 128.25, 129.27, 134.87, 142.47, 144.41, 145.32, 158.24, 161.04; MS *m*/*z* (%): 356 (M^+^, 27). Analysis calculated for C_16_H_16_N_6_S_2_ (356.09): C, 53.91; H, 4.52; N, 23.58; S, 17.99; Found: C, 53.80; H, 4.42; N, 23.39; S, 18.08%.

**4-Methyl-2-[((1-(thiophen-2-yl)ethylidene)hydrazineylidene]-5-((*p*-tolyldiazenyl)thiazol-3(2*H*)-amine (4b).** Red solid, 76% yield, m.p. 190–192 °C (EtOH); IR (KBr): *v* 3420, 3230 (NH_2_), 1598 (C=N) cm^−1^; ^1^H-NMR (DMSO-*d*_6_): *δ* 2.27 (s, 3H, CH_3_-thiazole),), 2.39 (s, 3H, Ar-CH_3_), 2.46 (s, 3H, CH_3_-C=N), 5.11 (s, 2H, Ar-H), 7.02–7.61 (m, 7H, Ar-H); ^13^C-NMR (DMSO-*d*_6_): *δ* = 12.89 (CH_3_-thiazole), 16.14 (CH_3_-C=N), 21.28 (Ar-CH_3_), 109.98, 125.13, 126.32, 127.19, 128.64, 130.27, 133.87, 140.47, 143.41, 145.32, 158.74, 160.81; MS *m*/*z* (%): 370 (M^+^, 39). Analysis calculated for C_17_H_18_N_6_S_2_ (370.10): C, 55.11; H, 4.90; N, 22.68; S, 17.31; Found: C, 55.03; H, 4.77; N, 22.51; S, 17.25%.

**5-((4-Chlorophenyl)diazenyl)-4-methyl-2-[((1-(thiophen-2-yl)ethylidene)hydrazineylidene] thiazol-3(2*H*)-amine (4c)**. Dark red solid, 75% yield, m.p. 205–207 °C (DMF); IR (KBr): *v* 3418, 3226 (NH_2_), 1600 (C=N) cm^−1^; ^1^H-NMR (DMSO-*d*_6_): *δ* = 2.28 (s, 3H, CH_3_-thiazole),), 2.46 (s, 3H, CH_3_-C=N), 5.03 (s, 2H, Ar-H), 7.08–7.63 (m, 7H, Ar-H); ^13^C-NMR (DMSO-*d*_6_): *δ* = 12.91 (CH_3_-thiazole), 16.04 (CH_3_-C=N), 111.18, 124.81, 125.94, 126.99, 127.53, 128.85, 132.81, 142.49, 143.41, 145.31, 158.17, 160.94; MS *m*/*z* (%): 392 (M^+^+ 2, 12), 390 (M^+^, 32). Analysis calculated for C_16_H_15_ClN_6_S_2_ (390.05): C, 49.16; H, 3.87; N, 21.50; S, 16.40; Found: C, 49.27; H, 3.71; N, 21.44; S, 16.52%.

**5-((2,4-Dichlorophenyl)diazenyl)-4-methyl-2-[((1-(thiophen-2-yl)ethylidene)hydrazineylidene] thiazol-3(2*H*)-amine (4d)**. Brown solid, 79% yield, m.p. 227–229 °C (DMF); IR (KBr): *v* 3415, 3228 (NH_2_), 1600 (C=N) cm^−1^; ^1^H-NMR (DMSO-*d*_6_): *δ* = 2.28 (s, 3H, CH_3_-thiazole),), 2.48 (s, 3H, CH_3_-C=N), 5.11 (s, 2H, Ar-H), 7.08–7.81 (m, 6H, Ar-H); ^13^C-NMR (DMSO-*d*_6_): *δ* = 13.27 (CH_3_-thiazole), 16.02 (CH_3_-C=N), 110.54, 124.91, 126.18, 127.89, 128.23, 130.27, 132.58, 134.87, 135.76, 142.43, 144.41, 145.65, 156.24, 162.14; MS *m*/*z* (%): 424 (M^+^, 25). Analysis calculated for C_16_H_14_Cl_2_N_6_S_2_ (424.01): C, 45.18; H, 3.32; N, 19.76; S, 15.07; Found: C, 45.03; H, 3.25; N, 19.68; S, 15.14%.

**3-Amino-2-[(1-(thiophen-2-yl)ethylidene)hydrazineylidene]-5-(2-(*p*-tolyl)hydrazineylidene) thiazolidin-4-one (8a)**. Yellow solid, 70% yield, m.p. 155–157 °C (EtOH); IR (KBr): *v* 3428, 3340, 3235 (NH_2_ & NH), 1680 (C=O), 1598 (C=N) cm^−1^; ^1^H-NMR (DMSO-*d*_6_): *δ* = 2.37 (s, 3H, Ar-CH_3_), 2.47 (s, 3H, CH_3_-C=N), 5.38 (s, 2H, NH_2_), 7.08–7.75 (m, 7H, Ar-H), 10.76 (s, 1H, NH); ^13^C-NMR (DMSO-*d*_6_): *δ* = 15.12 (CH_3_-C=N), 21.14 (Ar-CH_3_), 120.11, 122.13, 124.32, 125.19, 127.64, 128.55, 129.27, 140.47, 146.87, 153.41, 159.74 (Ar-Cs), 168.81 (C=O); MS *m*/*z* (%): 372 (M^+^, 62). Analysis calculated for C_16_H_16_N_6_OS_2_ (372.08): C, 51.60; H, 4.33; N, 22.56; S, 17.22; Found: C, 51.45; H, 4.19; N, 22.61; S, 17.35%.

**3-Amino-5-(2-(4-chlorophenyl)hydrazineylidene)-2-[(1-(thiophen-2-yl)ethylidene) hydrazineylidene]thiazolidin-4-one (8b)**. Yellow solid, 74% yield, m.p. 177–179 °C(DMF-EtOH); IR (KBr): *v* 3428, 3342, 3236 (NH_2_ & NH), 1681 (C=O), 1599 (C=N) cm^−1^; ^1^H-NMR (DMSO-*d*_6_): *δ* = 2.46 (s, 3H, CH_3_-C=N), 5.32 (s, 2H, NH_2_), 7.01–7.62 (m, 7H, Ar-H), 11.24 (s, 1H, NH); ^13^C-NMR (DMSO-*d*_6_): *δ* = 15.12 (CH_3_-C=N), 120.08, 121.93, 124.31, 125.21, 127.94, 128.51, 130.17, 141.41, 146.82, 151.87, 159.72 (Ar-Cs), 168.87 (C=O); MS *m*/*z* (%): 394 (M^+^+ 2, 6), 392 (M^+^, 16). Analysis calculated for C_15_H_13_ClN_6_OS_2_ (392.03): C, 45.86; H, 3.34; N, 21.39; S, 16.32; Found: C, 45.71; H, 3.45; N, 21.26; S, 16.44%.

**4-Phenyl-2-[((1-(thiophen-2-yl)ethylidene)hydrazineylidene]thiazol-3(2*H*)-amine (11a).** Yellowish-white crystals, 79% yield, m.p. 166–168 °C (EtOH); IR (KBr): *v* 3424, 3220 (NH_2_), 1599 (C=N) cm^−1^; ^1^H-NMR (DMSO-*d*_6_): *δ* = 2.37 (s, 3H, CH_3_-C=N), 4.77 (s, 2H, NH_2_), 7.02 (s, 1H, thiazole-H), 7.08–7.83 (m, 8H, Ar-H); ^13^C-NMR (DMSO-*d*_6_): *δ* = 15.12 (CH_3_-C=N), 112.08, 124.13, 125.31, 126.21, 127.94, 128.53, 135.17, 139.11, 144.54, 151.87, 159.72, 164.87 (Ar-Cs); MS *m*/*z* (%): 314 (M^+^, 51). Analysis calculated for C_15_H_14_N_4_S_2_ (314.07): C, 57.30; H, 4.49; N, 17.82; S, 20.39; Found: C, 57.16; H, 4.57; N, 17.69; S, 20.51%.

**2-[((1-(Thiophen-2-yl)ethylidene)hydrazineylidene]-4-(*p*-tolyl)thiazol-3(2*H*)-amine (11b).** Yellowish-white crystals, 75% yield, m.p. 149–151 °C (EtOH); IR (KBr): *v* 3430, 3214 (NH_2_), 1598 (C=N) cm^−1^; ^1^H-NMR (DMSO-*d*_6_): *δ* = 2.31 (s, 3H, Ar-CH_3_), 2.37 (s, 3H, CH_3_-C=N), 4.81 (s, 2H, NH_2_), 7.08 (s, 1H, thiazole-H), 7.23–7.79 (m, 7H, Ar-H); ^13^C-NMR (DMSO-*d*_6_): *δ* = 14.82 (CH_3_-C=N), 21.18 (Ar-CH_3_), 112.08, 124.04, 125.11, 126.18, 127.24, 128.53, 134.21, 139.11, 143.54, 152.87, 159.72, 163.94 (Ar-Cs); MS *m*/*z* (%): 328 (M^+^, 100). Analysis calculated for C_16_H_16_N_4_S_2_ (328.08): C, 58.51; H, 4.91; N, 17.06; S, 19.52; Found: C, 58.63; H, 4.80; N, 17.11; S, 19.63%.

**4-(4-Chlorophenyl)-2-[((1-(thiophen-2-yl)ethylidene)hydrazineylidene]thiazol-3(2*H*)-amine (11c)**. Yellow solid, 73% yield, m.p. 180–182 °C (DMF); IR (KBr): *v* 3428, 3220 (NH_2_), 1603 (C=N) cm^−1^; ^1^H-NMR (DMSO-*d*_6_): *δ* = 2.41 (s, 3H, CH_3_-C=N), 4.86 (s, 2H, NH_2_), 7.08 (s, 1H, thiazole-H), 7.11–7.84 (m, 7H, Ar-H); ^13^C-NMR (DMSO-*d*_6_): *δ* = 14.82 (CH_3_-C=N), 110.11, 120.13, 125.31, 126.21, 127.94, 128.53, 132.17, 135.11, 145.54, 149.87, 155.72, 164.87 (Ar-Cs); MS *m*/*z* (%): 350 (M^+^+ 2, 19), 348 (M^+^, 46). Analysis calculated for C_15_H_13_ClN_4_S_2_ (348.03): C, 51.64; H, 3.76; N, 16.06; S, 18.38; Found: C, 51.50; H, 3.84; N, 15.93; S, 18.44%.

**4-(4-Nitrophenyl)-2-[((1-(thiophen-2-yl)ethylidene)hydrazineylidene]thiazol-3(2*H*)-amine (11d)**. Brown solid, 81% yield, m.p. 194–196 °C (DMF); IR (KBr): *v* 3424, 3226 (NH_2_), 1600 (C=N) cm^−1^; ^1^H-NMR (DMSO-*d*_6_): *δ* = 2.43 (s, 3H, CH_3_-C=N), 4.84 (s, 2H, NH_2_), 7.05 (s, 1H, thiazole-H), 7.11–7.84 (m, 7H, Ar-H); ^13^C-NMR (DMSO-*d*_6_): *δ* = 14.59 (CH_3_-C=N), 110.11, 123.13, 124.31, 125.21, 126.94, 127.53, 142.17, 145.11, 149.54, 151.87, 155.72, 162.87 (Ar-Cs); MS *m*/*z* (%): 359 (M^+^, 73). Analysis calculated for C_15_H_13_N_5_O_2_S_2_ (359.05): C, 50.13; H, 3.65; N, 19.49; S, 17.84; Found: C, 50.05; H, 3.51; N, 19.37; S, 17.72%.

**1-[3-Amino-4-methyl-2-((1-(thiophen-2-yl)ethylidene)hydrazineylidene]-2,3-dihydrothiazol-5-yl)ethan-1-one (13a)**. Yellowish-white crystals, 77% yield, m.p. 156–158 °C (EtOH); IR (KBr): *v* 3416, 3220 (NH_2_), 1715 (C=O), 1591 (C=N) cm^−1^; ^1^H-NMR (DMSO-*d*_6_): *δ* = 2.25 (s, 3H, CH_3_-thiazole), 2.41 (s, 3H, CH_3_-C=N), 2.49 (s, 3H, COCH_3_), 5.12 (s, 2H, NH_2_), 7.02–7.62 (m, 3H, Ar-H); ^13^C-NMR (DMSO-*d*_6_): *δ* = 9.05 (CH_3_-thiazole), 15.37 (CH_3_-C=N), 40.52 (COCH_3_), 126.12, 127.49, 128.18, 142.11, 143.99, 145.66, 157.51, 164.47 (Ar-Cs), 181.91 (C=O); MS *m*/*z* (%): 294 (M^+^, 49). Analysis calculated for C_12_H_14_N_4_OS_2_ (294.06): C, 48.96; H, 4.79; N, 19.03; S, 21.78; Found: C, 48.79; H, 4.63; N, 19.01; S, 21.68%.

**Ethyl 3-amino-4-methyl-2-[((1-(thiophen-2-yl)ethylidene)hydrazineylidene]-2,3-dihydro thiazole-5-carboxylate (13b)**. Yellowish-white crystals, 71% yield, m.p. 141–143 °C (EtOH); IR (KBr): *v* 3420, 3222 (NH_2_), 1721 (C=O), 1595 (C=N) cm^−1^; ^1^H-NMR (DMSO-*d*_6_): *δ* = 1.14 (t, 3H, CH_3_-CH_2_), 2.28 (s, 3H, CH_3_-thiazole), 2.46 (s, 3H, CH_3_-C=N), 3.02 (q, 2H, CH_2_CH_3_), 5.11 (s, 2H, NH_2_), 7.02–7.62 (m, 3H, Ar-H); ^13^C-NMR (DMSO-*d*_6_): *δ* = 9.11 (CH_3_-thiazole), 15.05 (CH_3_-CH_2_), 15.36 (CH_3_-C=N), 56.52 (CH_2_CH_3_), 126.51, 127.45, 128.04, 143.11, 144.19, 145.68, 157.52, 163.47 (Ar-Cs), 182.61 (C=O); MS *m*/*z* (%): 324 (M^+^, 27). Analysis calculated for C_13_H_16_N_4_O_2_S_2_ (324.07): C, 48.13; H, 4.97; N, 17.27; S, 19.76; Found: C, 48.06; H, 4.83; N, 17.19; S, 19.61%.

#### 3.1.3. Alternate Synthesis of **4a** and **8a**

A mixture of *N*′-[1-(thiophen-2-yl)ethylidene]hydrazinecarbothiohydrazide (**5**) (0.214 g, 1 mmol) with the appropriate 2-oxo-*N*′-phenylpropanehydrazonoyl chloride (**3a**) or ethyl 2-chloro-2-(2-(*p*-tolyl)hydrazineylidene)acetate (**7a**) (1 mmol) in ethanol (20 mL) containing a catalytic amount of TEA was refluxed for 4 h (monitored by TLC). The precipitate formed was isolated by filtration, washed with methanol, dried, and recrystallized from EtOH to ensure the product was identical in all respects (m.p., mixed mp, and IR spectra) with the products **4a** or **8a,** respectively.

### 3.2. In Vitro Cytotoxic Activity

The cytotoxic potentials of the newly synthesized compounds was carried out at the Regional Center for Mycology and Biotechnology at Al-Azhar University, Cairo, Egypt. Cells were purchased from the Egyptian Holding Company for Biological Products and Vaccines (VACSERA, Giza, Egypt) and kept in a tissue culture unit. Cells were grown in Roswell Park Memorial Institute (RPMI)-1640 medium supplemented with 10% heat-inactivated fetal bovine serum (FBS), 50 units/mL penicillin, and 50 mg/mL streptomycin, and maintained in a humidified atmosphere containing 5% CO_2_ [11,25]. Cells were maintained as monolayer cultures using serial subculture. Cell culture reagents were obtained from Lonza (Basel, Switzerland). The anticancer activities of the rest of the compounds were evaluated in MCF-7 (breast cancer) cells. In addition, the sulforhodamine B (SRB) assay method, as described previously in [39,40], was used to determine cytotoxicity. Exponentially growing cells were collected using 0.25% trypsin–EDTA and seeded in 96-well plates at 1000–2000 cells/well in RBMI-1640 supplemented medium. After 24 h, cells were incubated for 72 h with various concentrations of the compounds tested. Following 72 h of incubation, the cells were fixed with 10% trichloroacetic acid for 1 h at 4 °C. Wells were stained for 10 min at room temperature with 0.4% sulforhodamine B (SRBC) dissolved in 1% acetic acid. Plates were air-dried for 24 h, and the dye was solubilized with Tris–HCl for 5 min on a shaker at 1600 rpm. The optical density (OD) of each well was measured spectrophotometrically at 564 nm using an ELISA microplate reader (ChroMate-4300, Palm City, FL, USA). IC_50_ values were calculated using a Boltzmann sigmoidal concentration–response curve using non-linear regression fitting models (Graph Pad, Prism Version 9, GraphPad Software, San Diego, CA, USA).

## 4. Conclusions

In summary, acetylthiophene was employed as a critical intermediate to synthesize a novel series of thiazole-bearing thiophene derivatives. The assigned structure for all of the newly synthesized compounds was elucidated by elemental and spectral analysis data, and the mechanisms accounting for their formation were discussed. The in vitro growth inhibitory activity of the synthesized compounds against MCF-7 tumor cells was investigated in comparison with cisplatin as a standard drug using the MTT assay, and the results revealed promising activities for compounds **4b** and **13a**. The results obtained from the computational studies, including molecular modeling, pharmacokinetics, and toxicity radar, supported the results obtained from the anticancer biological studies. 

## Data Availability

The data presented in this study are available on request from corresponding author.

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
