# Peer review of "Synthesis, Molecular Docking Study, and Cytotoxic Activity against MCF Cells of New Thiazole–Thiophene Scaffolds"

_molecules, 2022, doi:10.3390/molecules27144639_

Round 1
Reviewer 1 Report
This manuscript describes the synthesis of a series of thiazole bearing thiophene derivativess for the finding of potential anti-breast cancer agent. The in vitro growth inhibitory activity of the synthesized compounds against MCF-7 tumor cells were investigated using MTT assay. Compound 4b and 13a revealed promising activities. The computational studies, including molecular modeling, pharmacokinetics and toxicity radar, presented good explanation of the results obtained from anticancer biological studies. The results developed in this study should be of certain interest to medicinal chemists.
Minor revision is needed after the following issues are solved.
- The significance of the research should be addressed in the abstract and conclusion.
- The formats of the structure diagrams of the compounds in this paper are not uniform.
Author Response
- The significance of the research should be addressed in the abstract and conclusion.
Response: The abstract and conclusion have been updated
Appreciation for comment. A breast cancer scenario has been of high sound for many decades until now; the manuscript tried to introduce some novel drugs in the pharmaceutical industry, and in vitro studies can be translated into clinical studies later, which can serve as a potential breast cancer drug.
- The formats of the structure diagrams of the compounds in this paper are not uniform.
Response: The formats of the structure diagrams of the compounds were improved
Reviewer 2 Report
The manuscript , entitled “Synthesis, Anti-breast Cancer Assessment, and Molecular
Docking Study of some New Thiazole Clubbed Thiophene Scaffolds” deals with synthesis and evaluation of anti-breast cancer activity, as well as docking studies.Despite authors did quite a lot of job the discussion of results is missing.
1) Authors just gave an IC50 without discussing the results in comparison with control.
2) There is no discussion on structure-activity relationships.No correlation of experimental data with docking results.
3) Besides they did not gave the energy for cisplatin.
Page1. Line17 The following sentence needs revision.” In addition
, the bioactivities 17
of the synthesized compounds were evaluated their antitumor activities against MCF-7 tumor cells 18 compared to cisplatin reference drug-using MTT assay, and the results revealed promising activi- 19ties of compounds 4b and 13a
Page 1. Line 39.Should be: activity
Page 2. Line 46. The following sentence needs revision. Furthermore, the thiazole ring is 46
present in several anticancer drugs, such as Tiazofurin, Dasatinib, Dabrafenib, Bleomy- 47
cin, Ixabepilone, and Epothilone excellent pharmacological profiles, making this skeleton 48
an ideal candidate to develop more potent and safer anticancer drugs
Page 2. Line 61. Should be: hybrid structure that may lead
What were conditions to prepare compounds 6 from 5 as well as 8 from 7?
Page 4 Line 112. Should be: to obtain 11a-d
Page 4. Line 120. Should be: cisplatin as reference drug
Page 10. Line 158.Should be: experiments is poor ADME
References 14,18 , 21is not according to the journal rules.
There are 10 self-citation which is about 23% and it should be not more than 10%.
Author Response
- Authors just gave an IC50 without discussing the results in comparison with control.
Response: the results are now compared with IC50 for normal cell LLC-MK2 in the manuscript.
- There is no discussion on structure-activity relationships. No correlation of experimental data with docking results.
Response: Activity correlation and structure activities relationship is discussed and added
- Besides they did not gave the energy for cisplatin.
Response: cisplatin Energy is not calculated because it is difficult to be calculated using the software MOE2019. It appears in the MOE system as a square planner molecule, not a cis molecule. To solve this issue, we use Carboplatin as a refrance docking drug, and the new results are added in the corrected manuscript.
- Line17 The following sentence needs revision." In addition, the bioactivities of the synthesized compounds were evaluated their antitumor activities against MCF-7 tumor cells compared to cisplatin reference drug-using MTT assay, and the results revealed promising activities of compounds 4b and 13a
Response: It has been updated as “In addition, the bioactivities of the synthesized compounds were evaluated for their antitumor activities against MCF-7 tumor cells compared to a cisplatin reference drug and LLC-Mk2 normal cell line using MTT assay, and the results revealed promising activities of compounds 4b and 13a.”
- Page 1. Line 39. Should be: activity
Response: Done
- Page 2. Line 46. The following sentence needs revision. Furthermore, the thiazole ring is present in several anticancer drugs, such as Tiazofurin, Dasatinib, Dabrafenib, Bleomycin, Ixabepilone, and Epothilone excellent pharmacological profiles, making this skeleton an ideal candidate to develop more potent and safer anticancer drugs
Response: It has been updated as “Furthermore, the thiazole ring is present in several anticancer drugs (Figure 1), such as Tiazofurin, Dasatinib, Dabrafenib, Bleomycin, Ixabepilone, and Epothilone, making this skeleton an ideal candidate for developing more potent and safer anticancer drugs”
- Page 2. Line 61. Should be: hybrid structure that may lead
Response: Done
What were conditions to prepare compounds 6 from 5 as well as 8 from 7?
Response: EtOH\ TEA were used
- Page 4 Line 112. Should be: to obtain 11a-d
Response: Done
- Page 4. Line 120. Should be: cisplatin as a reference drug
Response: Done
- Page 10. Line 158. Should be: experiments is poor ADME
Response: Corrected and new ADME using Molinspiration is added with discussion
- References 14,18, 21 is not according to the journal rules.
Response: References 14,18, 21 were updated
- There are 10 self-citation which is about 23% and it should be not more than 10%.
Response: 3 references have been deleted ( references 25, 27 and 34) and the new refrences order has been updated
Reviewer 3 Report
In this work Gomha and coworkers present the development of the new thiazole bearing thiophene derivatives with potential anticancer activity. The manuscript contains the results of complementary experiments, is easy to follow and is potentially very interesting for the readers, but it needs to be revised by the Authors:
- In the introduction I suggest to present more information of molecular activity of compounds mentioned by Authors, especially 1,3-thiazole derivatives, including side effects
- The aim of the study should also clearly present the rationale for cell line based in vitro study
- I do not see the matching of the sentence in Lines 67-69 with the proposed studies on MCF7 cells;
- The Authors in cell based studies used the MTT assay. This assay allows to check the compounds effect on metabolic activity, therefore cytotoxicity – and this term is not equal to “anticancer effect”; I am aware that in many publications these terms are treated equally, but I suggest to be more precise and basing on the data precented in the article I suggest to modify and precise the title, since MTT assay does not correspond to “anti-breast cancer assessment”; it can be written as i.e. “Synthesis, Molecular Docking Study and Cytotoxic Activity on MCF cells of some New Thiazole Clubbed Thiophene Scaffolds”; also part 2.2 should be named rather as “Cytotoxic potential”/ “Cytotoxicity of selected compounds”. As cell model was used MCF7 cell line – have the Authors performed similar experiment with another cell line originated form breast cancer but with different pattern of estrogen receptors expression , i.e. MDA-MB-241, or maybe with any other cells, including primary cells or normal cell lines? This studies will allow to directly compare biological effect of obtained compounds.
- The Table 1 is not easy to follow – please modify it by presentation of each parameter only in one column; the Authors should comment the results presented in Table 1, for instance by explanation for choosing 4b and 13a for further studies.
- In cell based studies the Authors presented IC50 parameter, but there is no other experiment demonstrating the effect of compounds on other activities strongly connected with anticancer potential such as apoptosis induction, effect on MMP and ATP level, cell migration. These effects can be observed after cells treatment with concentration not exceeding IC50, but even at the highest non cytotoxic (IC0). Therefore I suggest to add to the Table 1 the data showing the IC0 concentration – this is the maximal concentration not influencing metabolic activity of cells.
- In cell-based in vitro experiments with chemically pure compounds usually the concentration is presented in µM/mM, therefore I strongly suggest to present the IC50 and IC0 in these unit instead of µg/mL, or just add this data as additional column in Table 1. What was the time of cell incubation with compounds?
- Have the Authors checked the synergic effect of synthesized compound and cisplatin with in vitro cell based studies? Since the molecular mechanism of these chemicals can differ, the mixture of compounds can be more effective than of corresponding single compounds
- Can the Authors predict what exact chemical modification can increase of the activity of compound?
- Please explain what exact two types of breast proteins were studied in docking simulation and the rationale for their choice basing on their significance in identified molecular cell signaling; there is also not sufficient information in regard to proteins studied in Table 2 description (there is data for one protein?, whereas in the text are mentioned two proteins?); Which protein was presented in Figure 4?
- in Table 2 there were “No measurable interaction” observed for 4a, 8a, 11b, 13a, 13b, however the Authors didn’t comment the significance of these results – should the compound interact or they shouldn’t with studied protein?
- What is the rationale to present the toxicity radar or predicted toxicity for 4a, 8a, 11b, 13a, 13b?
- In the methods there is no information of cell culture condition and cytotoxicity experiment (including information of cell density, stock solution concentration, solvent used for compounds, MTT assay procedure and calculation), molecular docking simulations and ADME determination.
- the conclusions should be modified in regard to presented above comments.
In summary I recommend the major revision of the manuscript.
Author Response
- In the introduction I suggest to present more information of molecular activity of compounds mentioned by Authors, especially 1,3-thiazole derivatives, including side effects
Response: Cancer is known as one of the main causes of death in the world; and many compounds have been synthesized to date with potential use in cancer therapy. Exploring new leads for identification of novel structures might be useful in designing less toxic selective and potent new anticancer agents and remains a major challenge for medicinal chemists.
Thus, designing and synthesizing more efficient agents with less adverse effects is essential. Thiazole is a versatile heterocycle, found in the structure of many drugs in use as well as anticancer agents.
The inadequate merits of available anticancer drugs for example lack of selectivity, target specificity, toxicity and developing resistance has led to serious consequences. Novel approaches targeting the specific molecular alterations that occur in tumour are being used in treatment and eradication of cancers. An ideal anticancer drug would eliminate cancer cells without harming normal tissues. Unfortunately, no currently available agents meet this criterion with a favourable therapeutic index. Therefore, novel anticancer agents owning target specific advanced mechanism of action with favourable therapeutic index are considered valuable in the field of oncology related drug discovery.
An increasing numbers of patents granted on thiazole-containing compounds and their exploration for possible clinical use in ongoing research and development reveal the potential of these derivatives in medicinal chemistry. Thus, it is clear that thiazole scaffold bearing compounds present plethora of applications in drug development for treatment against a variety of cancers. Despite remarkable advancements of thiazole containing compounds, still some concerted research endeavors are requisite;
- The aim of the study should also clearly present the rationale for cell line based in vitro study
Response Appreciation for comment. A breast cancer scenario has been of high sound for many decades until now; the manuscript tried to introduce some novel drugs in the pharmaceutical industry, and in vitro studies can be translated into clinical studies later, which can serve as a potential breast cancer drug.
- I do not see the matching of the sentence in Lines 67-69 with the proposed studies on MCF7 cells; The Authors in cell based studies used the MTT assay. This assay allows to check the compounds effect on metabolic activity, therefore cytotoxicity – and this term is not equal to "anticancer effect"; I am aware that in many publications these terms are treated equally, but I suggest to be more precise and basing on the data precented in the article I suggest to modify and precise the title, since MTT assay does not correspond to "anti-breast cancer assessment"; it can be written as i.e. "Synthesis, Molecular Docking Study and Cytotoxic Activity on MCF cells of some New Thiazole Clubbed Thiophene Scaffolds"; also part 2.2 should be named rather as "Cytotoxic potential"/"Cytotoxicity of selected compounds". As cell model was used MCF7 cell line – have the Authors performed similar experiment with another cell line originated form breast cancer but with different pattern of estrogen receptors expression , i.e. MDA-MB-241, or maybe with any other cells, including primary cells or normal cell lines? This studies will allow to directly compare biological effect of obtained compounds.
Response: the terms are changed according to the suggestion. Also, a test against normal cell line LLC-Mk2 was carried out; the SI" selectivity index was also calculated to compare results. All of that is added to the manuscript's new version.
The title of manuscript was updated as Reviewer suggession as follow:
“Synthesis, Molecular Docking Study and Cytotoxic Activity on MCF cells of some New Thiazole Clubbed Thiophene Scaffolds”
- The Table 1 is not easy to follow – please modify it by presentation of each parameter only in one column; the Authors should comment the results presented in Table 1, for instance by explanation for choosing 4b and 13a for further studies.
Response: The table is modified now in the new version of the manuscript
- In cell based studies the Authors presented IC50 parameter, but there is no other experiment demonstrating the effect of compounds on other activities strongly connected with anticancer potential such as apoptosis induction, effect on MMP and ATP level, cell migration. These effects can be observed after cells treatment with concentration not exceeding IC50, but even at the highest non cytotoxic (IC0). Therefore I suggest to add to the Table 1 the data showing the IC0 concentration – this is the maximal concentration not influencing metabolic activity of cells.
Response: Instead, tests against the LLC-Mk2 cell line were performed, and the table was changed while also SI was calculated. In addition, IC50 was calculated using Prism 9.4 model 2022
- In cell-based in vitro experiments with chemically pure compounds usually the concentration is presented in µM/mM, therefore I strongly suggest to present the IC50 and IC0 in these unit instead of µg/mL, or just add this data as additional column in Table 1. What was the time of cell incubation with compounds?
Response: Corrected to uM/mM
- Have the Authors checked the synergic effect of synthesized compound and cisplatin with in vitro cell based studies? Since the molecular mechanism of these chemicals can differ, the mixture of compounds can be more effective than of corresponding single compounds.
Response: No, reference for this study was added, and it carries no. [39]
Badisa, R.B.; Darling-Reed, S.F.; Joseph, P.; Cooperwood, J.S.; Latinwo, L.M.; Goodman, C.B. Selective cytotoxic activities of two novel synthetic drugs on human breast carcinoma MCF-7 Cells. Anticancer Res. 2009, 29, 2993–2996
- the Authors predict what exact chemical modification can increase of the activity of compound? Please explain what exact two types of breast proteins were studied in docking simulation and the rationale for their choice basing on their significance in identified molecular cell signaling; there is also not sufficient information in regard to proteins studied in Table 2 description (there is data for one protein?, whereas in the text are mentioned two proteins?); Which protein was presented in Figure 4?
Response: only 2W3L protein was used. It was chosen upon the literature reference [41] to represent why did we choose this protein.
Mashat, K.H.; Babgi, B.A.; Hussien, M.A.; Arshad, M.N.; Abdellattif, M.H. Synthesis, structures, DNA-binding and anticancer activities of some copper(I)-phosphine complexes. Polyhedron 2019, 158, 164–172. https://doi.org/10.1016/j.poly.2018.10.062
- in Table 2 there were "No measurable interaction" observed for 4a, 8a, 11b, 13a, 13b, however the Authors didn't comment the significance of these results – should the compound interact or they shouldn't with studied protein?
Response: comment is added now; no measurable interactions depend on the whole ligand exposure to the protein. It is also indicated that compound 4b was nearly the best one where IC50 is closer to cisplatin one and with good SI values of 46-30; these results are matched that 4b only which shows measurable interactions with high docking score energy.
- What is the rationale to present the toxicity radar or predicted toxicity for 4a, 8a, 11b, 13a, 13b?
Response: This choice was based upon the IC50 values obtained from biological studies.
- In the methods there is no information of cell culture condition and cytotoxicity experiment (including information of cell density, stock solution concentration, solvent used for compounds, MTT assay procedure and calculation), molecular docking simulations and ADME determination.
Response: All of them are corrected now in the manuscript
- the conclusions should be modified in regard to presented above comment
Response: we changed the concentrations upon suggestions
Round 2
Reviewer 2 Report
The manuscript , entitled “Synthesis, Anti-breast Cancer Assessment, and Molecular
Docking Study of some New Thiazole Clubbed Thiophene Scaffolds” after revision improved a little.But still they don’t cover two previous comments.
1) Authors just gave an IC50 without discussing the results in comparison with control.
2)There is no discussion on structure-activity relationships. A confusing information regarding docking.
Language needs improvement.
Lines 118-119. The following sentence needs revision” The cytotoxicity of the synthesized thiazoles 4a-d, 8a,b, 11a-d, and 13a,b were in-vestigated for their human breast cancer (MCF-7) cell line and normal cell line LLC-Mk2 using colorimetric MTT assay and cisplatin as reference drug.
Line 135.Should be: in the range of
Line 189. Should be: These results are matched that 4b is the only which shows measurable interactions with high docking score energy.
Line 196. Should be: -5.228 kcal/mol
Line 197. Should be: the only
In line 181 authors wrote “The docking score energy of the compounds 4a, 4b, 8a, 11b, 13a, 13b and CarboPt are (-5.911, -6.011,… and in line 195 they wrote” the docking score energy of compound 4b with PDB=1AO6 is -6.3 kcal/mol,
Also there is different energies for CarboPt
Line 221. Should be: phase of experiments.
Manuscript still have 21% self citation instead of 10%.
Author Response
Reviewer 2
Dear reviewer! Many thanks for your time spending and efforts to review the manuscript.
1. Authors just gave an IC50 without discussing the results in comparison with control.
Response: We performed tests against normal cell line, calculated SI, and discussed them (see Revised manuscript)
2. There is no discussion on structure-activity relationships. A confusing information regarding docking.
Response: SAR section was added
the structure-activity relationship is discussed in the Pred-hERG part with different compounds fig8, and fig. 9 showed Similar Off-target compounds of compounds 4b and 13a, respectively
3. Language needs improvement.
- Lines 118-119. The following sentence needs revision” The cytotoxicity of the synthesized thiazoles 4a-d, 8a,b, 11a-d, and 13a,b were in-vestigated for their human breast cancer (MCF-7) cell line and normal cell line LLC-Mk2 using colorimetric MTT assay and cisplatin as reference drug.
Response: Done
- Line 135.Should be: in the range of
Response: Done
- Line 189. Should be: These results are matched that 4b is the only which shows measurable interactions with high docking score energy.
Response: Done
- Line 196. Should be: -5.228 kcal/mol
Response: Done
- Line 197. Should be: the only
Response: Done
- In line 181 authors wrote “ The docking score energy of the compounds 4a, 4b, 8a, 11b, 13a, 13b and CarboPt are (-5.911, -6.011,… and in line 195 they wrote” the docking score energy of compound 4b with PDB=1AO6 is -6.3 kcal/mol, Also there are different energies for CarboPt
Response: After docking with 2W3L with 4b, we validate the results by docking with HSA (Humen serum albumin) protein 1AO6, and the result obtained from HSA protein is too neat from the results obtained from 2W3L.
Although different energies of CarboPt, we chose the higher one and wrote either with 2W3L or HAS to represent the results and their validation.
- Line 221. Should be: phase of experiments.
Response: Done
- Manuscript still has 21% self-citation instead of 10%.
Response: Another 4 references have been deleted
Reviewer 3 Report
The Authors answered to some of my questions, however the others concerns are not answered and the manuscript needs to be corrected.
I appreciate the additional data added to the manuscript, but since the Authors performed cytotoxicity studies and calculated IC50, therefore I strongly suggest to add the IC0 (the highest non cytotoxic concentration) value to the Table 1. Since methods of cell based study are poorly described please, add information of the studied concentration range of compounds used in cytotoxic studies (a representative graph for the most active compound can be presented).
Despite the information about performed correction of the concentration unit, the Authors did not change it within the text and in Tables.
Since the significant value of the presented data relies on the cell culture based studies I suggest to present the cell culture conditions in the manuscript, not as the supplementary assay. The manuscript is easy to follow, therefore I do not recommend presentation of this procedure in manuscript with lack of the details but only with the reference. Frankly speaking, the answer and correction is a bit sloppy… For example, in Supplementary material there is crystal violet staining procedure, whereas there are no data presented in the manuscript connected with cv; in the MTT assay are mentioned hepatocytes, whereas in the manuscript are mentioned MCF7 and LLC-MK2 cells. Still, there in the methods description there is lack of information of LLC-MK2 cell line source and culturing conditions, cell density, stock solution concentration of compounds and solvent used for dilution, no description of molecular docking simulations and ADME determination.
Additionally, there is still lack of the explanation why the docking studies were performed with the 2W3L protein – I suggest to find some information of this protein involvement in cell signalling and present them in the introduction part.
Author Response
Dear reviewer! Many thanks for your time spending and efforts to review the manuscript.
- I appreciate the additional data added to the manuscript, but since the Authors performed cytotoxicity studies and calculated IC50, therefore I strongly suggest to add the IC0 (the highest non cytotoxic concentration) value to the Table 1. Since methods of cell based study are poorly described please, add information of the studied concentration range of compounds used in cytotoxic studies (a representative graph for the most active compound can be presented).
Response: We performed tests against normal cell line, calculated SI, and discussed them
- Despite the information about performed correction of the concentration unit, the Authors did not change it within the text and in Tables.
Response: The concentration unit was changed from µg/mL into µM within the text and in Tables
Since the significant value of the presented data relies on the cell culture based studies I suggest to present the cell culture conditions in the manuscript, not as the supplementary assay. The manuscript is easy to follow, therefore I do not recommend presentation of this procedure in manuscript with lack of the details but only with the reference. Frankly speaking, the answer and correction is a bit sloppy… For example, in Supplementary material there is crystal violet staining procedure, whereas there are no data presented in the manuscript connected with cv; in the MTT assay are mentioned hepatocytes, whereas in the manuscript are mentioned MCF7 and LLC-MK2 cells.
Response: for cytotoxicity, the methods were added in the text after correction.
Still, there in the methods description there is lack of information of LLC-MK2 cell line source and culturing conditions, cell density, stock solution concentration of compounds and solvent used for dilution, no description of molecular docking simulations and ADME determination.
Response: Description of molecular docking is explained, and comparison validation with HSA was performed. ADME studies are explained (see Revised MS)
- Additionally, there is still lack of the explanation why the docking studies were performed with the 2W3L protein – I suggest to find some information of this protein involvement in cell signalling and present them in the introduction part.
Response: Apoptosis is an implicit cell suicide pathway that plays a pivotal role in normal and pathophysiological conditions. BCL-2 family proteins tightly regulate the intrinsic apoptotic pathway, and 2W3L is one of the proteins of this family; that is why we use it in our study"
The following paragraph was added into introduction section: “Apoptosis is essential to normal breast development and homeostasis. Pro-apoptotic and anti-apoptotic signals are tightly regulated in normal breast epithelial cells. Dysregulation of this balance is required for breast tumorigenesis and increases acquired resistance to treatments, including molecularly targeted therapies, radiation and chemotherapies. The pro-apoptotic or anti-apoptotic BCL-2 family proteins are key regulators of apoptosis process, and 2W3L is one of the proteins of this family. Therefore 2W3L is a promising target to improve breast cancer tumor cell killing [29, 30].
- Sathishkumar, N.; Sathiyamoorthy, S.; Ramya, M.; Yang, D.; Lee, H. N.; Yang, D. Molecular docking studies of anti-apoptotic BCL-2, BCL-XL, and MCL-1 proteins with ginsenosides from Panax ginseng, J. Enz. Inh.Med. Chem., 2012, 27, 685-692, DOI: 10.3109/14756366.2011.608663
- Williams, M.M.; Cook, R.S.Bcl-2 family proteins in breast development and cancer: could Mcl-1 targeting overcome therapeutic resistance? Oncotarget, 2015, 6, 3519-3530